# Strengthening of RC Structures by Using Engineered Cementitious Composites: A Review

**Xing-yan Shang [1] , Jiang-tao Yu [2], Ling-zhi Li [2,* and Zhou-dao Lu [2]**

[1]    College of Civil Engineering, Shandong Jianzhu University, Jinan 250101, China; 18818260950@126.com
[2]    College of Civil Engineering, Tongji University, Shanghai 200092, China; yujiangtao@tongji.edu.cn (J.-t.Y.);
    lzd@tongji.edu.cn (Z.-d.L.)
*    Correspondence: tjlilingzhi@gmail.com

**Abstract:** This paper presents a review of the recent work assessing the performance of building structures strengthened with engineered cementitious composite (ECC). ECC characterizes tensile strain hardening and multiple cracking properties, as well as strong interfacial bonding performance with substrate concrete, which makes it a promising retrofitting material. A lot of researches have been conducted on reinforced concrete (RC) structures, including beams, columns, beam–column joints, and fire-damaged slabs, strengthened with ECC material, and an extensive collection of valuable conclusions were obtained. These strengthening systems usually combine ECC with FRP textiles or steel bars to form a composite strengthening layer. The review demonstrates that ECC strengthening can greatly improve the performance of RC structures.

**Keywords:** ECC strengthening; bond strength; RC structures; FRP textile; steel bar

## 1. Introduction

Reinforced concrete (RC) structures may need strengthening or retrofitting due to material degradation, change of usage, and damages caused by earthquake and fire [1–3]. The most commonly used retrofitting materials for building structures were concrete, steel, and fiber reinforced polymers (FRPs) [4–6].

Besides concrete, steel is one of the most widely used materials in structural strengthening, and researchers have developed innovative strengthening strategies for the steel material. Wang et al. [5,7] proposed an innovative strategy to strengthen existing preloaded RC columns, in which precambered steel plates were bolted to the peripheral sides of RC columns to alleviate the shear lag effect. Lai et al. [8,9] proposed several innovative methods, such as steel rings, tie bars, spirals, and jackets, to strengthen the original and existing concrete-filled-steel-tube columns, which are highly effective since the confining stress of the core concrete can be increased. Su and Li [10–12] bolted steel plates to the side faces of RC beams, thus both the flexural and shear load capacity can be significantly enhanced without an obvious reduction of ductility.

FRP materials, including carbon, basalt, glass, and aramid FRPs, possess excellent properties, such as high tensile strength, high stiffness, corrosion resistance, and light weight. Due to the high efficiency, composite materials are widely used in engineering applications ranging from civil, aerospace, marine, etc. [13–15]. Many researches have been conducted on the performance of RC structures strengthened by bonded FRP composites. For example, [16] and [17] conducted experimental and theoretical studies on the flexural behavior of RC beams strengthened by FPRs. Triantafillou and Plevris [18,19] studied the time-dependent performance of RC beams retrofitted with FRP and established a systematic analysis procedure for the short-term flexural behavior of FRP-strengthened members. The authors of [20] and [21] investigated the performance of RC columns strengthened with

FRPs under various loading conditions. Chowdhury [22] and Williams [23] experimentally studied the fire performance of FRP reinforced concrete columns and slabs. Antonopoulos [24] and Yu [25] studied the seismic behavior of RC joints retrofitted with FRPs and found that FRP strengthening is a viable technique in increasing the strength, stiffness, and energy dissipation of RC joints. Steel plate is another commonly used retrofitting material. Su and Li [26,27] studied the performance of reinforced concrete beams strengthened with bolted steel plates. Hawileh RA et al. [28] used finite element software, ANSYS, to simulate the fire performance of RC T-beams strengthened with a CFRP layer. Barnes et al. [29] examined and compared the shear strengthening effect of concrete beams by attaching steel plates to the external surfaces from two methods, namely adhesive bonding and bolting. Zhang et al. [30] studied the performance and load capacity of RC slabs retrofitted with steel plates and found that bonded steel plates significantly enhanced the cracking load and ultimate load of the slabs.

Although there are many advantages for the strengthening techniques using FRP and steel plates, they also have several defects. For example, FRP is a brittle material and is prone to debonding from the substrate concrete. The external plates are susceptible to corrosion and the bolt holes will weaken the existing concrete to some extent. Therefore, new retrofitting techniques need to be developed for the strengthening of RC structures.

Engineered cementitious composite (ECC) is a kind of designed composite material reinforced with random short fibers with a volume fraction less than 2% to 3%, which features tensile strain hardening and multiple cracking properties [31]. The ultimate tensile strain can be more than 3%, even up to 8%, which is 300 to 800 times higher than that of normal or ordinary fiber reinforced concrete [32–36]. Cracking is one of the main reasons for the performance deterioration of material and then the structure; however, concrete is low in tension and is prone to cracking. Thus, researchers have been developing methods to reduce the width of cracks [37,38]. ECC is a material with a high crack-control capability. When ECC is applied in an RC beam, more tiny cracks spread over the tensile surface due to the bridging effect of fibers [39], while fewer wider cracks form on an ordinary RC beam [40]. Also, compared to FRP and steel, ECC is advantageous in terms of the material compatibility between the concrete substrate and the strengthening layer because it is a cement-based matrix.

These characters make it an attractive retrofitting material, thus many researchers have studied the strengthening effect of RC structures with ECC material.

## 2. Properties of ECC

Researchers have discovered many favorable properties of ECC, including satisfactory deformability, high energy absorption, delamination resistance, shear resistance, damage tolerance, and crack-width control, which make it an attractive choice in retrofitting building structures. Lim and Li [41] found that ECC was effective in trapping interface cracks so that surface spalling was eliminated, thus extending the service life of the structure. Also, Suthiwarapirak et al. [42] found that ECC has high resistance to fatigue. In addition, ECC has high resistance to freeze-thaw and delayed shrinkage cracking [43]. The crack-width-control capability also results in the self-healing property of ECC [44]. The small crack width can resist the migration of chlorides and other corrosive agents, and thus improve the durability of the materials [45]. Nehdi and Mohamed [46] proposed a novel hybrid ECC reinforced with short shape memory alloy (SMA) fibers and polyvinyl alcohol (PVA) fibers, and studied its impact response experimentally and numerically. Because of the shape memory effect of SMA fibers, the energy absorption capability of ECC increased after heat treatment, which is different from ordinary cementitious materials. Moreover, the concrete damage plasticity (CDP) and Drucker/Prager (DP) models were used to simulate the behavior of ECC subjected to impact loading.

The durability of ECC is superior because the transport velocity of water and chloride is much lower than that in ordinary concrete, attributed to the special crack pattern of ECC, which is closely spaced multiple cracks. Liu et al. [47] found that ECC maintains high mechanical properties under sulfate and combine sulfate-chloride conditions. Also, the use of limestone powder had positive effects on the durability of ECC because of its fine particle structure, leading to good fiber dispersion [48].

ECC incorporated with steel and polypropylene (PP) fibers improved plastic shrinkage performance and showed low chloride penetration, indicating superior durability [49]. Moreover, the durability of ECC material is expected to increase the durability of structures in which ECC is applied, such as RC structures strengthened with ECC materials.

Despite the many advantages of ECC, the drying shrinkage in ECC is a major problem because of the lack of coarse aggregate included to provide internal restraints. However, many researchers proposed various solutions to solve this problem, such as increasing the sand: binder ratio, fly ash content, and the addition of some additives or agents [50]. Zhang et al. [51] applied pre-wetted calcined zeolite particles in high strength ECC and the shrinkage was reduced by more than 60%. Calcium sulfoaluminate (CSA)-based expansive additive (EXA) [52] and rice husk ash (RHA) [53] were incorporated into the ECC mixture to reduce the shrinkage of ECC and resist cracking.

## 3. The Bond Performance between ECC and Concrete

Although ECC has many attractive characteristics, a strong interfacial bond is needed when it is applied to strengthen RC structures. The bonding behavior between ECC and concrete directly influences the strengthening effects. Thus, some researchers investigated the bond strength between ECC and concrete [54–59]. The experimental methods used to investigate the interfacial bonding include the slant shear test, direct shear test, splitting tensile test, and pull-out test. The interfacial bond strength between ECC and concrete in the references are summarized in Table 1, where the slant shear strength is found to be much higher than the rest indexes. The ECC type has little effect on the bond strength. The bond strength between S_ECC (ECC with slag) and concrete is a little higher than that between F_ECC (ECC with class-F fly ash) and concrete. However, the bond strength between ECC (S_ECC or F_ECC) and concrete increases a lot compared with MSC (microsilica concrete). Surface treatment played a great role on the bond strength, i.e., specimens treated with sandblast showed the highest bond strength both in the slant shear test and splitting tensile test. However, silane applied between ECC and concrete has almost no influence on the bond strength. The bond strength increased with the surface roughness. Also, Lim and Li [59] studied the interfacial fracture behavior of the infrastructures repaired with ECC.

**Table 1.** Summary of the interfacial bond strength between ECC and concrete.

| Reference | Specimen Series Number | Bond Strength/ MPa | | | | Details |
|---|---|---|---|---|---|---|
| | | Slant Shear Test | Direct Shear Test | Splitting Tensile Test | Pull-out Test | |
| Sahmaran et al. [54] | F_ECC | 21.7 | | 3.39 | | ECC with Class-F fly ash |
| | S_ECC | 24.3 | | 3.59 | | ECC with slag |
| | MSC | 15.6 | | 3.04 | | Microsilica concrete |
| Zhu et al. [55] | S-1 | | 2.132 | | | No silane treatment |
| | S-2 | | 2.164 | | | With silane treatment |
| | P-1 | | | | 1.283 | No silane treatment |
| | P-2 | | | | 1.039 | With silane treatment |
| Tayeh et al. [56] | AC | 8.68 | | 1.85 | | As cast surface treatment |
| | G | 13.92 | | 3.24 | | Grooved surface treatment |
| | DH | 12.27 | | 2.60 | | Drill hole treatment |
| | SB | 17.81 | | 3.79 | | Sandblast treatment |
| | WB | 12.75 | | 2.96 | | Wire brush treatment |
| Wang [57] | A-I | | 2.55 | 1.37 | 2.18 | Wire brushing surface |
| | A-II | | 3.17 | 3.06 | 2.46 | Artificial chiseling surface with low surface roughness |
| | A-III | | 3.68 | 4.71 | 2.74 | Artificial chiseling with high surface roughness |

## 4. RC Structures Strengthened with ECC

Due to the low tensile capacity of concrete, RC structures are vulnerable to cracking. Fortunately, the high tensile strain capacity of ECC can compensate this defect of concrete, thus being an ideal retrofitting material. Some researchers have investigated the application of ECC strengthening in RC structures, including RC beams, columns, and beam–column joints. ECC is usually applied in combination with FRP textiles (Figure 1) or steel bars in order to significantly increase the strength of the structure member. When ECC is introduced in a structure member, multiple tiny cracks form on the tensile surfaces or areas under complex stresses, such as the beam bottom soffit and joint core area, while the load bearing capacity of the structure member will not decrease. This multiple cracking process usually results in a ductile failure mode. FRP textiles and steel bars can also improve the structure integrity to some extent, especially in the strengthening of RC columns. However, the main role of FRP textiles or steel bars is to increase the strength of the structure members.

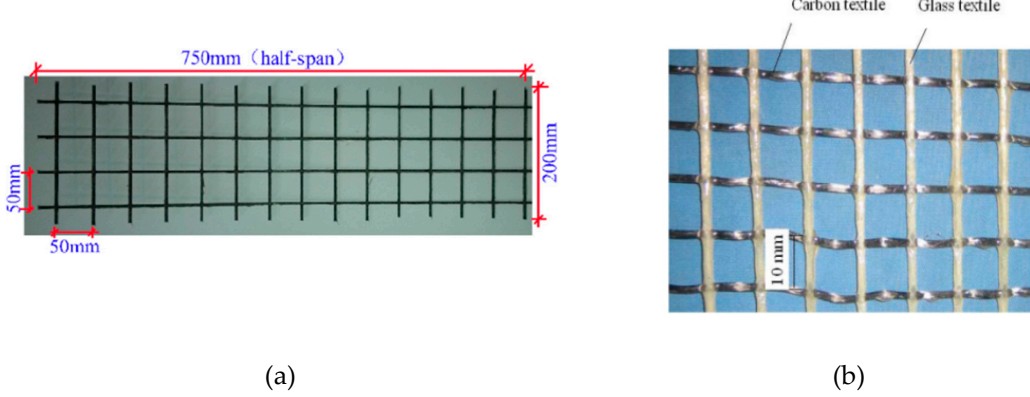

(a) (b)

**Figure 1.** FRP textile. (**a**) BFRP grid [60], (**b**) Composite textiles [61].

### 4.1. RC Beams Strengthened with ECC

There are two strengthening arrangements for RC beams, i.e., the strengthening layer attached on the bottom soffit for bending strengthening (Figure 2a) and U-shaped jackets used for shear strengthening (Figure 2b). The manufacturing process of beams strengthened with ECC and high strength reinforcing steel (HSRS) bars is shown in Figure 3. For beams strengthened with ECC and FRP textile, the process is similar to those with ECC only, except that after a layer of ECC is plastered on the original concrete substrate, FRP bonding is conducted.

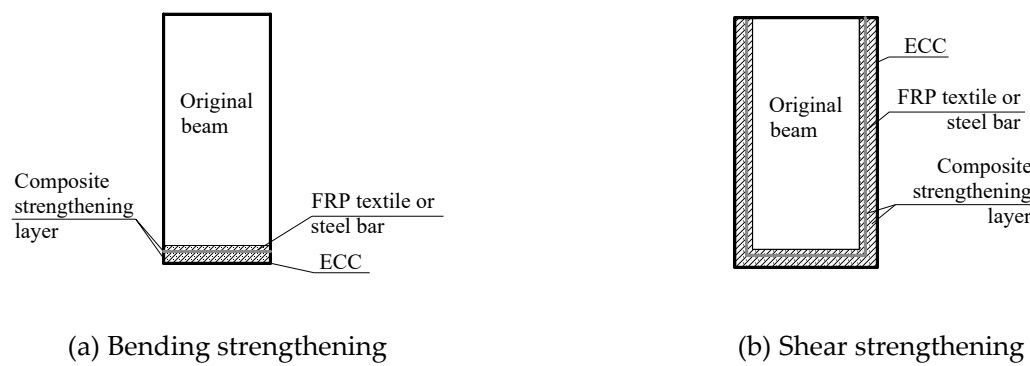

(a) Bending strengthening (b) Shear strengthening

**Figure 2.** Two strengthening schemes.

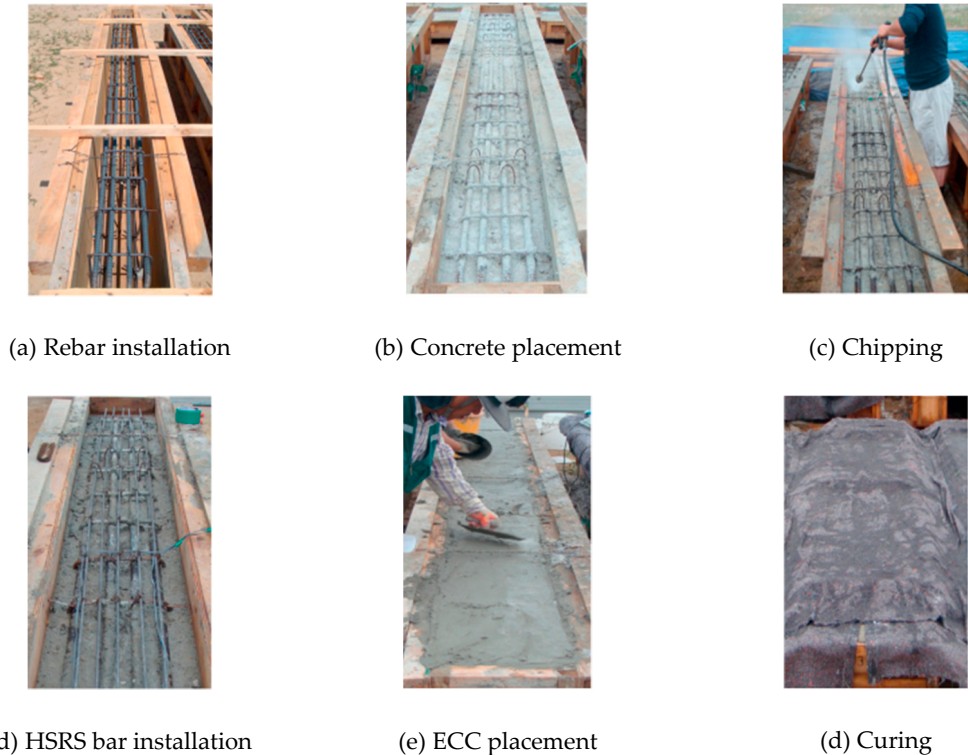

(a) Rebar installation          (b) Concrete placement          (c) Chipping

(d) HSRS bar installation       (e) ECC placement               (d) Curing

**Figure 3.** Manufacturing process for beam strengthening (pictures from Kim [62]).

Zheng et al. [60] combined ECC and a basalt FRP (BFRP) grid (Figure 1a) to strengthen RC beams, which were externally attached on the beam bottom soffit and worked together as a composite reinforcement layer (CRL), as shown in Figure 4. The test parameters were the grid thickness and the ratio of the CRL length to the clear span of the beam. The beams were subjected to four-point bending tests to investigate their flexural behavior. The strengthened beams, BB-1-500, BB-3-500, BB-3-450, and BB-3-400, failed in bending with BFRP rupture and concrete crushing, while the strengthened beam, BB-5-500, failed with partial debonding of the strengthening layer and BFRP rupture (Figure 5a). Multiple fine cracks occurred in the composite reinforcement layer. However, for beams strengthened with BFRP only [63], BFRP was completely debonded from the substrate concrete (Figure 5b). Compared with beams strengthened with BFRP only, we can see the effective interface bonding of beams strengthened with ECC and BFRP, which can also be confirmed by analyzing the strain distribution along the beam depth, where the results showed no slip between the concrete and strengthening layer. The specimen details and strengthening effects of the beams are listed in Table 2, where it is evident that the combination of ECC and BFRP grid strengthening greatly increased the strength and stiffness of the beams; however, the displacement was decreased.

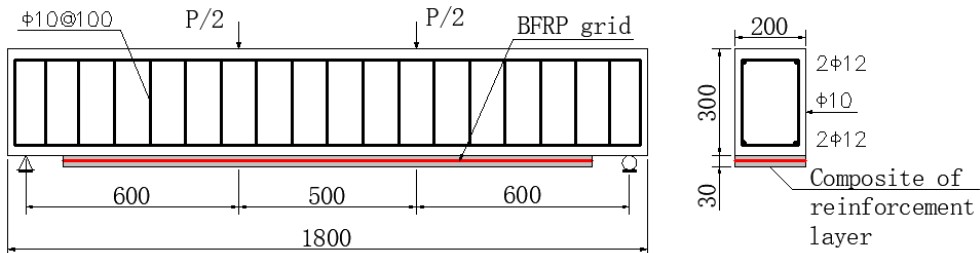

**Figure 4.** Details of strengthened beams (adapted from [60]).

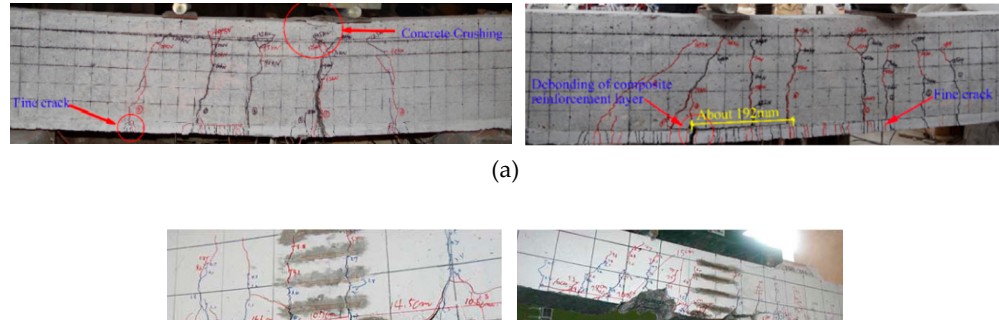

(a)

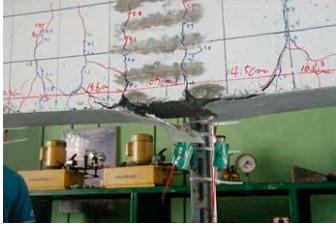 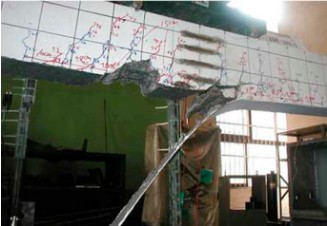

(b)

**Figure 5.** Failure modes of RC beams strengthened with (**a**) CRL strengthened beams BB-1-500 and BB-5-500 (pictures from Zheng, et al. [60]) and (**b**) BFRP strengthened beams B10-L1 and B8-L3 (pictures from Sim, et al. [63]).

**Table 2.** Specimen design and strengthening effect [60].

| Specimen | Thickness of BFRP Grid /mm | CRL Length/Clear Span | Ultimate Load /kN | Increase /% | Displacement /mm | Increase /% | Failure Mode |
|---|---|---|---|---|---|---|---|
| CL | - | - | 120 | - | 25.88 | | Flexural failure with concrete crushing |
| BB-1-500 | 1 | 0.88 | 129 | 7.5 | 10.32 | −60.1 | Flexural failure with BFRP rupture and concrete crushing |
| BB-3-500 | 3 | 0.88 | 139 | 15.8 | 10.51 | −59.4 | Same as BB-1-500 |
| BB-5-500 | 5 | 0.88 | 151 | 25.8 | 10.54 | −59.3 | Flexural failure with BFRP rupture and CRL partial debonding |
| BB-3-450 | 3 | 0.82 | 138 | 15.0 | 9.80 | −62.1 | Same as BB-1-500 |
| BB-3-400 | 3 | 0.76 | 136 | 13.3 | 11.00 | −57.5 | Same as BB-1-500 |

Similarly, Dai et al. [61] used textile reinforced ECC (TR-ECC) to strengthen RC beams. The textile used was carbon textile and glass textile (Figure 1b), different from the BFRP grid in [60]. In this paper, the beams were strengthened with an ECC strengthening layer attached on the bottom soffit except one (*ECC), whose ECC was attached on both the bottom soffit and the two vertical surfaces. The strengthening effect of ECC was also compared with that of steel fiber reinforced concrete (SFRC) and ordinary mortar. The test parameters included the textile stiffness, the matrix type, and the anchorage length of the textiles. Test results revealed that TR-ECC strengthening significantly increased the flexural capacity and stiffness of the beams. Beams strengthened with TR-ECC showed better performance than those with SFRC and mortar. Similar to [60], debonding was not found before failure, indicating an effective bond between the strengthening layer and concrete substrate. Also, Yang et al. [64] researched the flexural performance of RC beams strengthened with ECC and CFRP, which is proven as an effective strengthening scheme.

Kim et al. [62] investigated the flexural behavior of RC beams retrofitted with strain-hardening cementitious composite (SHCC, i.e., ECC) and HSRS bars. The strengthening layer was attached on the bottom soffits of the beams, and then flexural bending tests were conducted. The test parameter was the number of HSRS bars. The test results revealed that ECC combined with HSRS bars can control the crack width as well as increase the stiffness and load bearing capacity of beams. Moreover, the stiffness and load bearing capacity increased with the number of HSRS bars.

Huang and Chen [65] manufactured six RC cantilever beams, five of which were shear strengthened with mortar jackets or ECC jackets, as shown in Figure 6. The jackets were reinforced with or without steel meshes. The parameters were the matrix, layer of meshes, and category of meshes, as shown in Table 3. All the beams were subjected to vertical cyclic loading. Experimental test results indicated that the ECC jacket was more effective than mortar jackets in restraining concrete spalling and crushing. Also, the combination of the ECC jacket and steel meshes was prone to the promotion of multiple tiny cracks and further enhancement of the shear performance of beams, including the shear strength capacity, deformability, and pinching behavior. In addition, the beam strengthened with the ECC jacket and a layer of steel mesh was the most effective retrofitting scheme.

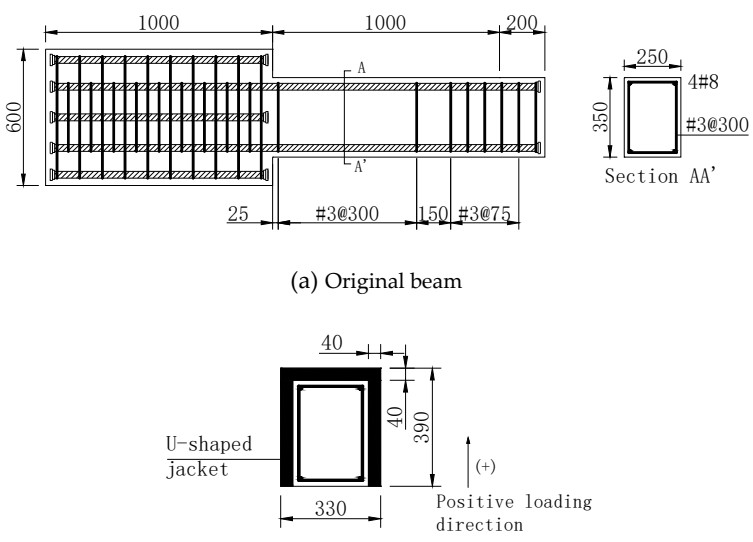

(a) Original beam

(b) Cross section of the retrofitted beam

**Figure 6.** Beam design details (adapted from [65]).

**Table 3.** Specimen details and strengthening effects.

| Specimen Number | Matrix | Reinforcement | Stiffness/kN/m | Strength/kN | Maximum Drift/% | Failure Mode |
|---|---|---|---|---|---|---|
| B-CONT | - | - | 10.4 (+) | 130.3 (+) | 3 | Flexural-shear |
| | | | 11.3 (−) | 130.8 (−) | | |
| B-E | ECC | 0 | 13.1 (+) | 143.8 (+) | 5 | Flexural-shear |
| | | | 14.3 (−) | 142.8 (−) | | |
| B-M-6 | Mortar | 6 layers of wire meshes | 15.9 (+) | 151.3 (+) | 7 | Flexure |
| | | | 15.0 (−) | 138.8 (−) | | |
| B-M-1 | Mortar | 1 layer of bar meshes | 15.3 (+) | 176.2 (+) | 7 | Flexure |
| | | | 14.9 (−) | 165.2 (−) | | |
| B-E-6 | ECC | 6 layers of wire meshes | 13.9 (+) | 158.3 (+) | 7 | Flexural-shear |
| | | | 13.0 (−) | 160.3 (−) | | |
| B-E-1 | ECC | 1 layer of bar mesh | 15.9 (+) | 167.2 (+) | 8 | Flexure |
| | | | 16.8 (−) | 164.7 (−) | | |

Shang et al. [66] investigated the shear behavior of fire damaged RC beams strengthened with steel reinforced ECC. Two types of ECC, including PVA-ECC and PE-ECC (polyethylene, PE), were used to retrofit the damaged beams. Test results showed that the strength, stiffness, and displacement increased a lot compared with the control beams, and the beams strengthened with PE-ECC showed better behavior.

Maalej and Leong [67] compared the behavior of RC beams strengthened with externally bonded FPR and FRP incorporating ECC, in which the concrete in the bottom region of the beams was replaced by ECC, as shown in Figure 7. Beams A1 and A2 were the control beam and FRP bonded beam, respectively. Beam ECC-1 was strengthened by ECC only, and beam ECC-2 was strengthened by FRP incorporating ECC. It is obvious that FRP bonding significantly increased the bearing capacity. However, the deflection was decreased, owing to FRP debonding. When ECC was incorporated in ECC-2, the deflection capability increased considerably compared with A2 due to the tensile ductility of ECC materials but was still less than the control beam, A1. Finite element software, DIANA, was used to simulate the behavior of the beams, for instance, the FRP strains and load–deformation behavior. The simulation results agreed well with the experimental results. As shown in Figure 8, the maximum CFRP stress in the strengthened beam with bonded FRP only was much higher than that with FRP incorporating ECC. Therefore, the load capacity of RC beams strengthened with the latter strategy was much higher. In addition, ECC delayed the CFRP debonding, thus increasing the effectiveness of the CFRP material. Meanwhile, ECC also reduced the loss of deflection capacity caused by FRP debonding or fracture.

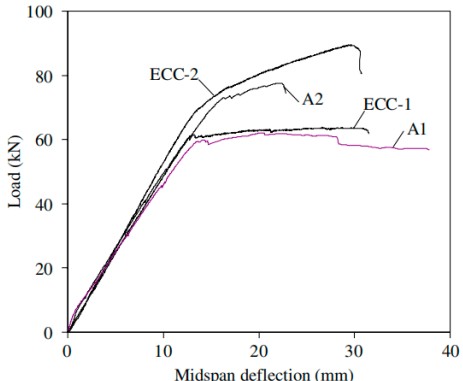

**Figure 7.** Load–deflection curves [67].

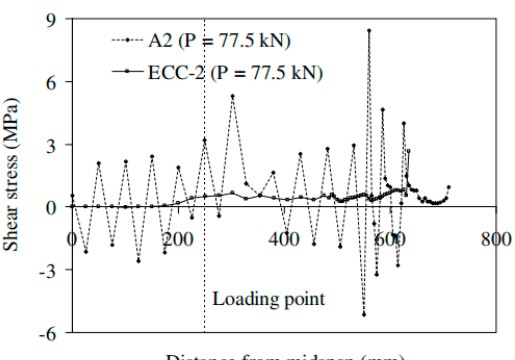

**Figure 8.** CFRP stress distribution [67].

Researchers also proposed a formula to predict the bearing capacity of the flexural strengthened beams. Based on the plane cross-section assumption and force equilibrium, flexural capacity formula were proposed according to different failure modes in [64]:

$$M_{\mathrm{u}} = \begin{cases} \alpha_1 f_{\mathrm{c}} x_{\mathrm{f}} b (d'_{\mathrm{s}} - \frac{x_{\mathrm{f}}}{2}) + A_{\mathrm{s}} f_{\mathrm{y}} (h_0 - d'_{\mathrm{s}}) + (A_{\mathrm{f}} f_{\mathrm{fu}} + A_{\mathrm{E}} f_{\mathrm{e}})(h_{\mathrm{f}} - d'_{\mathrm{s}}), & (0 \le x_{\mathrm{f}} < x_{\mathrm{bf}}) \\ \alpha_1 f_{\mathrm{c}} x_{\mathrm{f}} b (h_{\mathrm{f}} - \frac{x_{\mathrm{f}}}{2}) + A'_{\mathrm{s}} \sigma'_{\mathrm{s}} (h_{\mathrm{f}} - d'_{\mathrm{s}}) - A_{\mathrm{s}} f_{\mathrm{y}} (h_{\mathrm{f}} - h_0), & (x_{\mathrm{bf}} < x_{\mathrm{f}} < x_{\mathrm{lfc}}) \\ \alpha_1 f_{\mathrm{c}} x_{\mathrm{f}} b (h_{\mathrm{f}} - \frac{x_{\mathrm{f}}}{2}) + A'_{\mathrm{s}} f'_{\mathrm{y}} (h_{\mathrm{f}} - d'_{\mathrm{s}}) - A_{\mathrm{s}} f_{\mathrm{y}} (h_{\mathrm{f}} - h_0), & (x_{\mathrm{lfc}} < x_{\mathrm{f}} < x_{\mathrm{uf}}) \\ \alpha_1 f_{\mathrm{c}} x_{\mathrm{f}} b (h_{\mathrm{f}} - \frac{x_{\mathrm{f}}}{2}) + A'_{\mathrm{s}} f'_{\mathrm{y}} (h_{\mathrm{f}} - d'_{\mathrm{s}}) - A_{\mathrm{s}} \sigma_{\mathrm{s}} (h_{\mathrm{f}} - h_0), & (x_{\mathrm{f}} \ge x_{\mathrm{uf}}) \end{cases} \quad (1)$$

where $x_f$ is the depth of the equivalent rectangular compressive stress zone; $x_{bf}$, $x_{lfc}$, and $x_{uf}$ are the depths of the compressive stress zone corresponding to different failure modes; $h_f$, $h_0$, and $d'_s$ are the distances from the centroids of CFRP, tensile steel bars, and compressive steel bars to the top of concrete compressive zone, respectively; $A_s$ ($A'_s$), $\sigma_s$ ($\sigma'_s$), and $f_y$ ($f'_y$) are the cross-section area, the stress, and the yield strength of the tensile (or compressive) steel bars, respectively; $A_f$ ($A_E$) and $f_{fu}$ ($f_e$) are the cross-section area and the tensile strength of the FRP textile and ECC matrix, respectively; $\alpha_1$ is a coefficient; and $f_c$ is the compressive strength of concrete.

　　There is no formula for the shear strengthening of RC beams as yet, which needs further study.

### 4.2. RC Columns Strengthened with ECC

　　ECC combined with FRP textiles or steel bars was also applied in the strengthening of RC columns, including circular and square ones, or to change the section shape of the columns after strengthening, as shown in Figure 9. In this strengthening scheme, the main role of the strengthening layer was to confine the core concrete, which was thus under a triaxial compression stress state. Therefore, the compressive strength and the ultimate compressive strain of the confined concrete increased. ECC can delay the formation and propagation of cracks and improve the brittle failure mode of concrete columns, thus increasing the energy absorption capability.

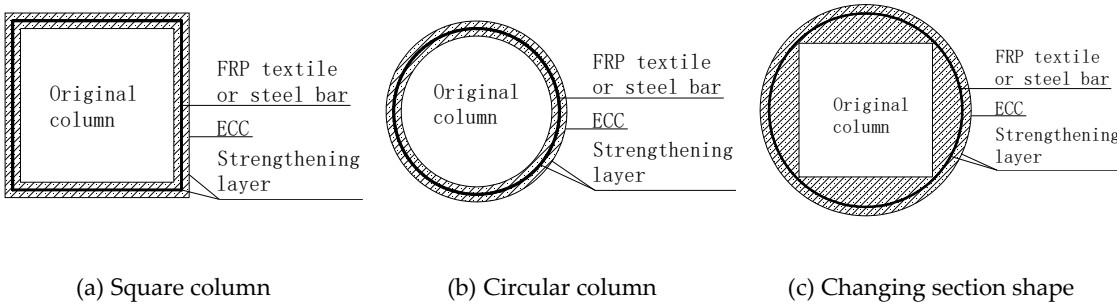

(a) Square column　　　　　　(b) Circular column　　　　　(c) Changing section shape

**Figure 9.** Strengthening schemes for RC columns.

　　Four RC short columns were strengthened by Deng et al. [68] with ECC jackets, while another was strengthened with a ferro-cement jacket for control. The retrofitting details are shown in Figure 10. All specimens were subjected to lateral cyclic loading, which simulated seismic excitation. The effects of different design schemes on the performance of RC short columns were investigated. Test results showed that the failure modes of columns strengthened with ECC jackets were much more ductile than those of the control specimens, as shown in Table 4. The shear strength and deformation capacity of the RC columns were effectively improved.

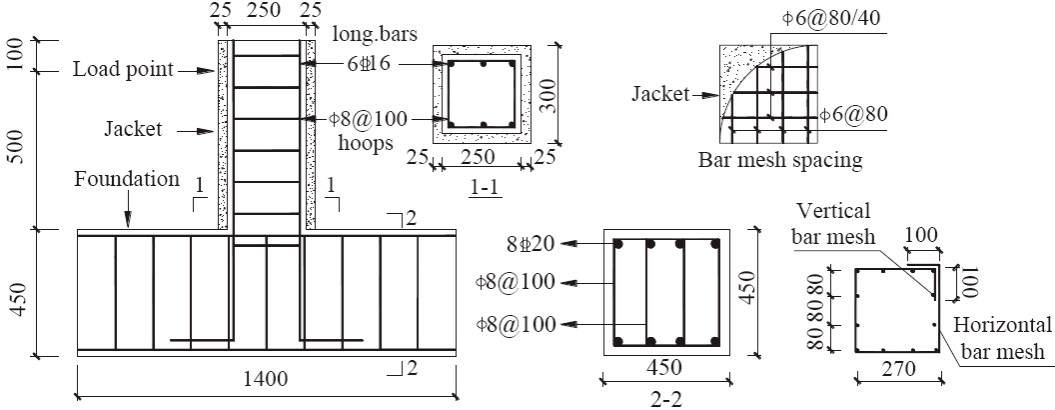

**Figure 10.** Retrofitting details of RC short columns [68].

**Table 4.** Specimen details and strengthening effect.

| Specimen Number | Matrix | Reinforcement | Design Axial Load Ratio | Tested Axial Load Ratio | Maximum Load/kN | Displacement /mm | Failure Mode |
|---|---|---|---|---|---|---|---|
| C-1 | - | - | 0.80 | 0.35 | 212.43 | 6.65 | Shear failure |
| C-2 | Mortar | 80 × 80 mm Ferro-cement | 0.47 | 0.22 | 317.3 | 5.36 | Brittle flexural-shear failure |
| C-3 | ECC | 0 | 0.47 | 0.21 | 309.11 | 7.43 | Ductile flexural-shear failure |
| C-4 | ECC | 80 × 80 mm bar mesh | 0.47 | 0.21 | 318.55 | 5.51 | Ductile flexural-shear failure |
| C-5 | - | - | 1.00 | 0.43 | 229.32 | 4.91 | Shear failure |
| C-6 | ECC | 80 × 80 mm bar mesh | 0.58 | 0.26 | 319.43 | 6.31 | Ductile flexural-shear failure |
| C-7 | ECC | 40 × 80 mm bar mesh | 0.58 | 0.26 | 316.3 | 5.69 | More ductile failure mode |

Zhu and Wang [69] investigated the loading and deformation capacities of RC circular columns strengthened with FRP textile and ECC (Figure 11). The specimens were subjected to a compressive test. The experimental results revealed that all strengthened RC columns failed by FRP rupture (Table 5). The load and deformation capacity of confined RC columns increased with the reinforced FRP layer. Besides, the composite strengthened layer combined with FRP textile and ECC can provide effective lateral confining stress to the retrofitted RC columns and delay the yielding of the longitudinal bar.

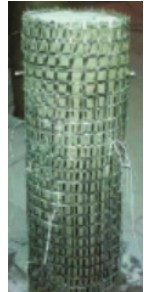 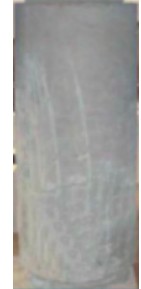

(**a**) Confined BFRP textile    (**b**) ECC casting

**Figure 11.** Retrofitting details of RC circular columns [69].

**Table 5.** Specimen details and test results.

| Specimen Number | Matrix | Number of Textile | Ultimate Load /kN | Ultimate Displacement /mm | Failure Mode |
|---|---|---|---|---|---|
| RC | - | - | 785.8 | 1.24 | Longitudinal bar yield and concrete crushing |
| FRE1 | ECC1 | 1 | 1076.0 | 2.17 | Longitudinal bar yield and BFRP rupture |
| FRE1E | ECC1 | 1 | 1277.8 | 2.12 | Same as FRE1 |
| FRE2E | ECC1 | 2 | 1136.2 | 2.23 | Same as FRE1 |
| FRE3E | ECC1 | 3 | 1171.0 | 2.38 | Same as FRE1 |
| FRSE1 | ECC2 | 1 | 1229.4 | 2.37 | Same as FRE1 |

AL-Gemeel and Zhuge [70] investigated the strengthening effect of basalt fiber textile combined with ECC to confine RC columns. The effect of ordinary FRP strengthening decreased owing to the square shape of the column and brittleness of FRP, so that ductile ECC was incorporated and a circular mold was adopted to overcome these drawbacks. The cross section of the column changed from a square to circular shape. The strengthening details of the confined RC columns are shown in Figure 12. The test parameters were the matrix type and basalt textile spacing, as shown in Table 6. Compressive

strength tests were conducted on all the columns. Test results showed that this strengthening scheme significantly increased the bearing capacity and ductility of RC columns.

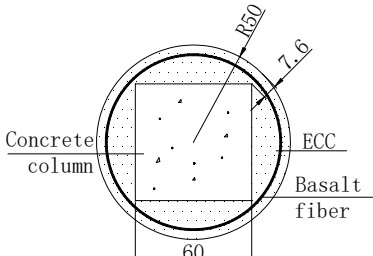

**Figure 12.** Retrofitting details of RC square columns changed to a circular shape (adapted from [70]).

**Table 6.** Specimen details and strengthening effect.

| Specimen Number | Matrix | Basalt Textile Spacing/mm | Strength /MPa | Axial Strain /μm/m | Hoop Strain /μm/m | Failure Mode |
|---|---|---|---|---|---|---|
| C | - | - | 20.2 | 576 | −118 | Brittle failure |
| CC-E | ECC | 0 | 38.9 | 3090.9 | −1027.8 | PVA fiber slipped |
| CC-EB10 | ECC | 10 | 31.2 | 2662.1 | −1222.5 | Basalt textile fractured |
| CC-EB25 | ECC | 25 | 35.8 | 2807.1 | −870.9 | Basalt textile fractured |
| CC-EB50 | ECC | 50 | 32.2 | 2117.9 | −575.4 | Basalt textile fractured |
| CC-MB25 | Cement mortar | 25 | 28.5 | 1067.2 | −195.7 | Fragment delamination |

Researchers also proposed several formulae to predict the shear capacity of the strengthened RC columns [68]:

$$V_{\mathrm{mc}} = V_{\mathrm{C}} + V_{\mathrm{L}}, \tag{2}$$

$$V_{\mathrm{C}} = \frac{1.75}{\lambda + 1} f_{\mathrm{t}} b h_0 + f_{\mathrm{yv}} \frac{A_{\mathrm{sv}}}{s} h_0 + 0.07 N, \tag{3}$$

$$V_{\mathrm{L}} = \alpha_{\mathrm{c}} \frac{1.75}{\lambda + 1} f_{\mathrm{tl}} A_{\mathrm{l}} + \alpha_{\mathrm{s}} f_{\mathrm{yvl}} \frac{A_{\mathrm{svl}}}{s_{\mathrm{l}}} h_{0\mathrm{l}}, \tag{4}$$

where $V_{\mathrm{C}}$ ($V_{\mathrm{L}}$) and $h_0$ ($h_{\mathrm{l}}$) are the shear contributions and the effective heights of the original RC column (the strengthening layer), respectively; $0.07N$ is the shear contribution of the axial load to the shear strength; $\lambda$ is the shear span ratio of the columns; $f_{\mathrm{t}}$ and $f_{\mathrm{tl}}$ are the tensile strength of concrete and ECC, respectively; $f_{\mathrm{yv}}$ ($f_{\mathrm{yvl}}$), $s$ ($s_{\mathrm{l}}$), and $A_{\mathrm{sv}}$ ($A_{\mathrm{svl}}$) are the yield strength, the spacing, and the cross-section area of the stirrups (the FRP textiles or steel bars), respectively; and $\alpha_{\mathrm{c}}$ and $\alpha_{\mathrm{s}}$ are the effective strength coefficients of ECC and FRP textiles, respectively.

*4.3. RC Beam–Column Joints Strengthened with ECC*

The ductility of ECC combined with the high tensile strength of FRP textile can lead to a high toughness strengthening scheme. The formation of multiple tiny cracks, which open and close under reversal cyclic load, results in satisfactory energy dissipation. Esmaeeli et al. [60] investigated the effectiveness of strengthening RC beam–column joints with ECC and CFRP laminates. The concrete cover in the joint was replaced by ECC. Two different strengthening schemes were used: (1) Two-sided retrofitting system, i.e., applying ECC and CFRP to the front and rear faces of the joints; and (2)

four-sided retrofitting system, i.e., applying ECC and CFRP to all sides of the elements of the joints. Figure 13 shows a retrofitted joint. The joints were subjected to lateral displacement-controlled cyclical loads. Based on the test results, the hysteretic response, dissipated energy, and displacement ductility were analyzed. It was found that the two-sided retrofitting system could only restore the lateral load bearing and energy dissipation capacities while the four-sided retrofitting system significantly increased the load bearing and energy dissipation capacities.

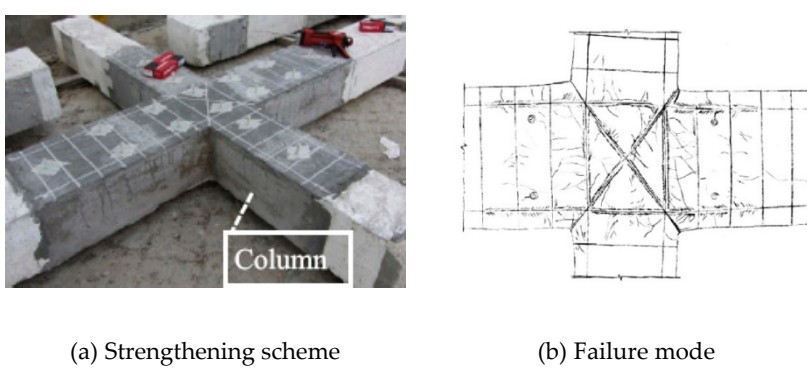

(a) Strengthening scheme                    (b) Failure mode

**Figure 13.** Retrofitted joints [71].

### 4.4. Fire Damaged RC Slabs Strengthened with ECC

Gao et al. [72] and Hu et al. [73] used BFRP reinforced ECC to strengthen fire damaged RC slabs. The strengthening procedure is shown in Figure 14. The test parameters included the fire exposure time, matrix type, and layers of BFRP grids, as shown in Table 7. This strengthening technique significantly increased the load-carrying capacity of RC slabs. Compared with polymer modified mortar, the ECC application resulted in a higher load capacity and ductility. However, the BFRP layer increased the load capacity but decreased the ductility at the same time. Most of the slabs failed in bending with the yielding of longitudinal bars, except slab B1-3, which failed in shear with plate end debonding of BFRP. This is because the BFRP debonding weakened the end section of the plate and led to shear failure.

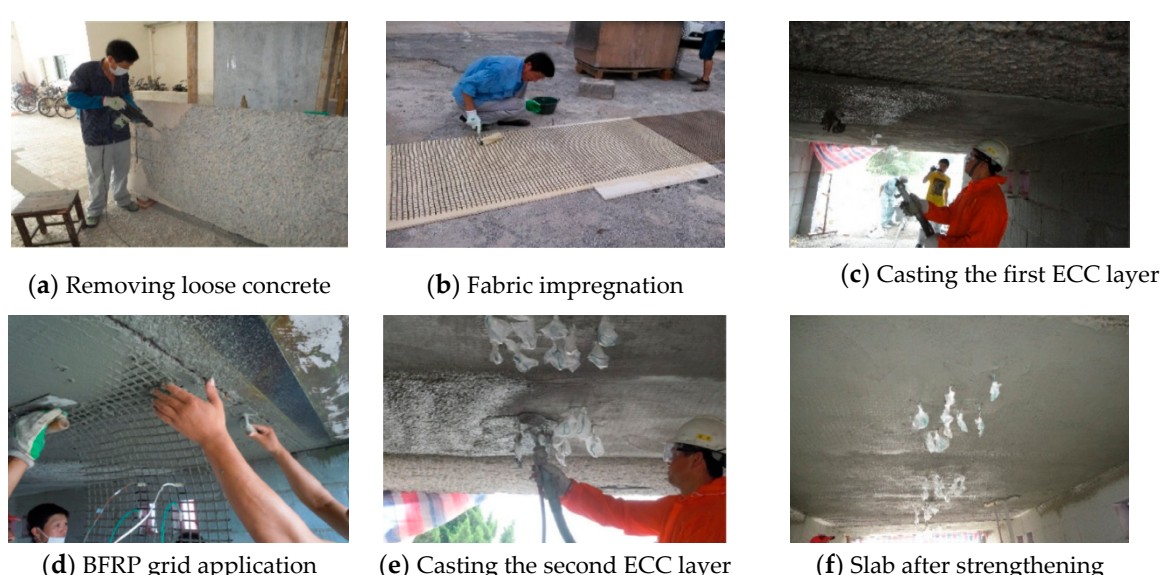

(**a**) Removing loose concrete    (**b**) Fabric impregnation    (**c**) Casting the first ECC layer

(**d**) BFRP grid application    (**e**) Casting the second ECC layer    (**f**) Slab after strengthening

**Figure 14.** Strengthening procedures for fire damaged RC slabs (pictures from Gao et al. [72]).

**Table 7.** Test parameters and results.

| Specimen Number | Fire Exposure Time/min | Matrix | Layers of BFRP Grid | Plate Thickness after Strengthening/mm | Peak Load /kN | Ultimate Displacement /mm | Ductility Factor | Failure Mode |
|---|---|---|---|---|---|---|---|---|
| B0 | - | - | - | 100 | 27.3 | | - | Steel yielding and large deflection, flexural failure |
| B1-0 | 60 | - | - | 100 | 26.1 | | - | Steel yielding and large deflection, flexural failure |
| B1-1 | 60 | Collapsed | | - | - | - | - | - |
| B1-2 | 60 | PMM | 3 | 137 | 48.3 | 117.9 | 3.65 | Steel yielding and BFRP rupture, flexural failure |
| B1-3 | 60 | ECC | 3 | 126 | 53.3 | 90.5 | 2.81 | Steel yielding, BFRP debonding and shear failure |
| B2-0 | 81 | - | - | 100 | 21.2 | - | - | Steel yielding and large deflection, flexural failure |
| B2-1 | 81 | PMM | 2 | 125 | 35.8 | 108.4 | 3.96 | Steel yielding and BFRP rupture, flexural failure |
| B2-2 | 81 | PMM | 3 | 134 | 50.1 | 120.8 | 3.54 | Steel yielding and BFRP rupture, flexural failure |
| B2-3 | 81 | ECC | 3 | 126 | 62.2 | 139.8 | 4.06 | Steel yielding and BFRP rupture, flexural failure |

Some researchers [74–77] also applied ECC materials to strengthen unreinforced masonry walls, which were constructed using bricks and mortars to resist horizontal loads, such as earthquakes. ECC strips or coatings, with or without the combination of steel mesh, were used to strengthen the unreinforced masonry walls and investigated the performance of the strengthened walls subjected to different loads, i.e., uniformly-distributed load, patch load, and low-velocity projectile impact. These efforts found that ECC retrofitting could significantly improve the lateral strength, ductility, and energy dissipation of unreinforced masonry walls.

Table 8 summarizes the strengthening materials and strengthening effects, from which we can see that for RC beams flexural-strengthened with ECC and FRP textiles, the strength and stiffness both increased; however, the deformation ductility decreased due to the brittleness of FRP textiles. Comparatively, beams strengthened with ECC and steel bars or steel meshes showed better behavior in ductility. The strengthening methods improved almost all the mechanical properties of RC columns, joints, and masonry structures.

**Table 8.** ECC strengthening summary.

| Strengthened Member Type | Reference | Strengthening Material | Strength | Stiffness | Deformation or Ductility | Energy Dissipation |
|---|---|---|---|---|---|---|
| RC beam | Zheng et al. [60] | ECC + BFRP grids | Increased | Increased | Decreased | - |
| | Dai et al. [61] | ECC + carbon or glass textile | Increased | Increased | Decreased | - |
| | Kim et al. [62] | ECC + high strength steel bar | Increased | Increased | No increase | - |
| | Yang et al. [64] | ECC+CFRP grid | Increased | Increased | Decreased | - |
| | Hung and Chen [65] | ECC + steel mesh | Increased | Increased | Increased | Increased |
| | Shang et al. [66] | ECC + steel bar | Increased | Increased | Increased | - |
| | Maalej and Leong [67] | ECC + CFRP | Increased | Increased | Decreased | - |
| RC column | Deng et al. [68] | ECC + bar mesh | Increased | Increased | Increased | Increased |
| | Zhu and Wang [69] | ECC + BFRP textile | Increased | Increased | Increased | - |
| | AL-Gemeel and Zhuge [70] | ECC + BFRP textile | Increased | - | Increased | - |
| RC beam-column joints | Esmaeeli et al. [71] | ECC + CFRP laminate | Increased | - | Increased | Increased |
| Fire damaged slabs | Gao et al. [72] and Hu et al. [73] | ECC + BFRP textile | Increased | - | Increased | - |
| Masonry buildings | Maalej et al. [74] | ECC + steel mesh | Increased | Increased | Increased | - |
| | Deng and Yang [75] | ECC | Increased | Increased | Increased | Increased |
| | Lin et al. [76] | ECC | Increased | No increase | Increased | - |
| | Lin et al. [77] | ECC | Increased | - | - | - |

ECC has been used in the real world. For example, it was used to retrofit the Ten-nou JR tunnel to prevent the concrete lining from cracking and spalling [78]. The concrete dams of Hohenwarte II power plant were repaired with ECC to prevent the leakage of water [79].

## 5. Conclusions

In reviewing the structural strengthening applications of ECC, it is clear that the strain hardening characteristic and strong interfacial bonding strength with concrete make it an ideal material for structural strengthening. Several points are drawn from this review:

(1)　The tensile strain hardening property of ECC overcomes the drawback of concrete, which is low in tension and prone to cracking. Thus, ECC is expected to be a promising retrofitting material.

(2)　The bond strengths between ECC and concrete, including interfacial tensile strength and interfacial shear strength, are strong enough to transfer force from an original RC structure to the strengthening layer. The bond strength in the slant shear test is obviously higher than other tests. Also, the bond strength increases with the surface roughness.

(3)　ECC combined with FRP textiles or steel bars has been used to strengthen RC structures. This strengthening technique can significantly increase the load capacity, stiffness, deformation capacity, and energy dissipation, except for several bending strengthening beams, the displacement of which was decreased.

(4)　Calculation formulae for the flexural capacity of RC beams and the shear capacity of RC columns retrofitted with this composite layer have been proposed. However, the calculation formula for the shear capacity of strengthened RC beams has not been found as yet.

The durability of ECC materials have been studied by some researchers and the durability is expected to increase the long-term performance of strengthened RC structures. However, the durability of ECC strengthened RC structures have not been researched yet. Also, the bond performances between ECC and concrete after being subjected to high temperatures, including the interfacial tensile strength, interfacial shear strength, and interfacial fracture behavior, have not been studied at present. The formulae for the shear capacity of RC beams also need further study.

**Author Contributions:** Conceptualization, L.-z.L.; Writing-Original Draft Preparation, X.-y.S.; Writing-Review & Editing, L.-z.L.; Supervision, J.-t.Y., Z.-d.L.; Founding Acquisition, L.-z.L., J.-t.Y.

**Funding:** This study was funded by National Natural Science Foundation of China (Project No. 51778496, No. 51778497), Doctoral Foundation of Shandong Jianzhu University (Project No. XNBS1849) and foundation from Cooperated Innovation Center of Shandong Province for disaster prevention and reduction of civil engineering structure (Project No. XTP201921).

**Conflicts of Interest:** The authors declare no conflict of interest.

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
