# Peer review of "Strengthening of RC Structures by Using Engineered Cementitious Composites: A Review"

_sustainability, doi:10.3390/su11123384_

Round 1
Reviewer 1 Report
This paper reviews novel works on the use of engineered cementitious composites to retrofit and strengthen concrete and masonry structures. This review is timely and of interest to this community. This review requires revisions in the form of the following items.
Major Questions, Comments and Concerns
1) A more in-depth discussion on the differences between ECCs and traditional bonded FRPs systems is needed to pinpoint the advantageous of ECCs.
2) The review could utilize few papers that deal with modeling the FRPs and ECCs systems. The mention and discussion of this aspect can help complete this review. Some of these articles may include the following, or the authors may option to use other references.
a. https://doi.org/10.1016/j.engstruct.2009.08.008
b. https://doi.org/10.3390/app9050994
3) The authors are encouraged to add a new section towards the end of this review to highlight research needs and future trends within this area.
Author Response
Dear Editor,
Thank you very much for your valuable advice. We have revised the paper according to your kind comments.
Questions:
A more in-depth discussion on the differences between ECCs and traditional bonded FRPs systems is needed to pinpoint the advantageous of ECCs.
Reply:
We have discussed the differences between ECCs and traditional bonded FRPs in detail in the revised manuscript. Please check lines 202-221 in the revised manuscript, which are highlighted in yellow.
Questions:
The review could utilize few papers that deal with modeling the FRPs and ECCs systems. The mention and discussion of this aspect can help complete this review. Some of these articles may include the following, or the authors may option to use other references.
a. https://doi.org/10.1016/j.engstruct.2009.08.008
b. https://doi.org/10.3390/app9050994
Reply:
We discussed and cited several papers that deal with modeling the FRPs and ECCs systems. Please check lines 38-40, 72-77, and 210-218 in the revised manuscript.
Questions:
The authors are encouraged to add a new section towards the end of this review to highlight research needs and future trends within this area.
Reply:
We have added a future studies section in lines 348-353 in the revised manuscript.
Thank you again!
Best wishes!
Xingyan Shang, Jiangtao Yu, Lingzhi Li and Zhoudao Lu.

Reviewer 2 Report
The paper presents a review of the recent work researching on assessing the performance of building structures strengthened with engineered cementitious composite. Some general points that need to be addressed by the authors to enhance the quality of the manuscript are reported below in the section “general comments”.
The paper is a good contribution to the field. In what follows, I list some comments and suggestions that shall be addressed by the authors while finalizing the manuscript in a revision process.
General comments:
1) Introduction section is well written; However, I strongly recommend to extend the initial part by introducing a preliminary discussion (just 2 rows i.e. “Due the high efficiency, composite materials are widely used in several engineering applications ranging from civil, aerospace, marine etc. (XX-XX).” ) about the composites material and they use in other engineering applications. To frame this introductive part, I recommend to consider the following paper in which composites are used in others frameworks:
· A numerical model based on ALE formulation to predict crack propagation in sandwich structures. Frattura Ed Integrita Strutturale, 13(47), 277-293. doi:10.3221/IGF-ESIS.47.21
· Bond between carbon fabric-reinforced cementitious matrix and masonry substrate. Journal of Materials in Civil Engineering, 31(1) doi:10.1061/(ASCE)MT.1943-5533.0002561
· An interface approach based on moving mesh and cohesive modeling in Z-pinned composite laminates. Composites Part B: Engineering, 135, 207-217. doi:10.1016/j.compositesb.2017.10.018
2) Figures 4 6, 10 seem to be in bad quality, I recommend to replace them.
Author Response
Dear Editor,
Thank you very much for your valuable advice. We have revised the paper according to your kind comments.
Questions:
Introduction section is well written; However, I strongly recommend to extend the initial part by introducing a preliminary discussion (just 2 rows i.e. “Due the high efficiency, composite materials are widely used in several engineering applications ranging from civil, aerospace, marine etc. (XX-XX).” ) about the composites material and they use in other engineering applications. To frame this introductive part, I recommend to consider the following paper in which composites are used in others frameworks:
· A numerical model based on ALE formulation to predict crack propagation in sandwich structures. Frattura Ed Integrita Strutturale, 13(47), 277-293. doi:10.3221/IGF-ESIS.47.21
· Bond between carbon fabric-reinforced cementitious matrix and masonry substrate. Journal of Materials in Civil Engineering, 31(1) doi:10.1061/(ASCE)MT.1943-5533.0002561
· An interface approach based on moving mesh and cohesive modeling in Z-pinned composite laminates. Composites Part B: Engineering, 135, 207-217. doi:10.1016/j.compositesb.2017.10.018
Reply:
We have accepted your valuable recommendation and extended the initial part of the introduction section. Please check lines 25-26 in the revised manuscript, which is highlighted in yellow.
Questions: Figures 4, 6, 10 seem to be in bad quality, I recommend to replace them.
Reply: We have replaced these figures by clear ones. Please check lines 153, 190-193,and 279 in the revised manuscript, which is highlighted in yellow.
Thank you again!
Best wishes!
Xingyan Shang, Jiangtao Yu, Lingzhi Li and Zhoudao Lu.
Reviewer 3 Report
The authors presented a review of the current-of-state practices for ECC-based structural retrofit. In general, the manuscript is not good written and particularly poor organized. Although the ECC is a great solution for concrete strengthening or updating, the authors provided a very limited view of the work associated with ECC.
The current format of the manuscript is not acceptable, and the authors have to provide a more comprehensive review for ECC, not just including the pros but also cons of this material. Since ECC has high shrinkage issues, the authors should review the associated issues to give a better view for the readers. Also, the most papers cited in the manuscript demonstrated that ECC has to combine with other reinforcement (mesh, FRP bar/grid), while the authors overlook the contributions of those reinforcement, whether those provide tension as well?
The second paragraph in the introduction was over painted, or not reasons to lead to why we need ECC. The authors should rephrase those statements carefully.
The major concern is that, this is Sustainability Journal, and the authors should address the structural updating associated with long-term durability, as well as mechanical enhancement. It seems the authors just focused on the mechanical improvement, which is not sufficient. Also, whether are there the real-world applications that could demonstrate the cases?
Author Response
Dear Editor,
Thank you very much for your valuable advice. We have revised the paper according to your kind comments.
Questions:
The current format of the manuscript is not acceptable, and the authors have to provide a more comprehensive review for ECC, not just including the pros but also cons of this material. Since ECC has high shrinkage issues, the authors should review the associated issues to give a better view for the readers.
Reply:
We have discussed the drawbacks of ECC materials, especially the high shrinkage problems. The corresponding solutions proposed by some researchers have also been discussed. Please check lines 87-93 in the revised manuscript, which are highlighted in yellow.
Questions:
The most papers cited in the manuscript demonstrated that ECC has to combine with other reinforcement (mesh, FRP bar/grid), while the authors overlook the contributions of those reinforcement, whether those provide tension as well?
Reply:
We have discussed the contribution provided by FRP or mesh reinforcement. Please check lines 122-124, 204-208 in the revised manuscript.
Questions:
The second paragraph in the introduction was over painted, or not reasons to lead to why we need ECC. The authors should rephrase those statements carefully.
Reply:
We have revised the second paragraph in the introduction. Please check lines 45-47.
Questions:
The major concern is that, this is Sustainability Journal, and the authors should address the structural updating associated with long-term durability, as well as mechanical enhancement. It seems the authors just focused on the mechanical improvement, which is not sufficient. Also, whether are there the real-world applications that could demonstrate the cases?
Reply:
The durability of ECC materials have been studied by researchers and we have added this part. please check lines 78-86.
However, the durability of RC structures strengthened with ECC has not found at jet and it is the trend of future research, which has been pointed out in lines 351-352 and 370-371.
ECC has been applied in the real-world and several examples are discussed in lines 345-347 in the revised manuscript.
Thank you again!
Best wishes!
Xingyan Shang, Jiangtao Yu, Lingzhi Li and Zhoudao Lu.

Reviewer 4 Report
It is an interesting and original paper. In this work, the main problems related to the strengthening of RC structures by using engineered cementitious composites were investigated. The paper perfectly fits with the aims and scope of the Sustainability Journal. However, it is recommended that the paper be revised before publication. The specific amendments are as follows:
(1) Figures 3, 5 and 12 show numerous pictures from the tests. However, not specified who is their author (Authors of the paper or someone else). Please, complete this.
(2) Section 3 (Table 1) lists the ECC parameters with the addition of fly ash. Please specify what type of fly ash, i.e. siliceous or calcareous.
(3) Figure 10 is difficult to read. Please correct it.
(4) In section 5, please provide the main conclusions from the research (preferably at several points). Please clearly indicate the originality of the presented solutions and the practical benefits of using them.
(5) In the paper in many places mentioned advantages of the presented method, such as increased resistance to cracking of composite and reducing the size of the microcracks in its structure. At this point, other modern methods to reduce the size of cracks in the material should also be presented. One of them is the modification of concrete composites with active mineral additives. Therefore, the following articles should be discussed and cited:
“The influence of microcrack width on the mechanical parameters in concrete with the addition of fly ash: Consideration of technological and ecological benefits”, Construction and Building Materials, 2019.
“An assessment of microcracks in the Interfacial Transition Zone of durable concrete composites with fly ash additives”, Composite Structures, 2018.
Author Response
Dear Editor,
Thank you very much for your valuable advice. We have revised the paper according to your kind comments.
Questions:
Figures 3, 5 and 12 show numerous pictures from the tests. However, not specified who is their author (Authors of the paper or someone else). Please, complete this.
Reply:
We have specified the authors of the pictures. Please check lines 134, 157, 159 and 327 in the revised manuscript, which are highlighted in yellow.
Questions:
Section 3 (Table 1) lists the ECC parameters with the addition of fly ash. Please specify what type of fly ash, i.e. siliceous or calcareous.
Reply:
The fly ash in Table 1 is Class-F fly ash. Please check line 104 and Table 1 in the revised manuscript.
Questions:
Figure 10 is difficult to read. Please correct it.
Reply:
The figure name has been revised. The shape of the strengthened column has also been explained in the above text. Please check lines 272-275 and line 280 in the revised manuscript.
Questions:
In section 5, please provide the main conclusions from the research (preferably at several points). Please clearly indicate the originality of the presented solutions and the practical benefits of using them.
Reply:
The conclusion section has been revised and 5 points are drawn. Please check lines 357-374 in the revised manuscript.
Questions:
In the paper in many places mentioned advantages of the presented method, such as increased resistance to cracking of composite and reducing the size of the microcracks in its structure. At this point, other modern methods to reduce the size of cracks in the material should also be presented. One of them is the modification of concrete composites with active mineral additives. Therefore, the following articles should be discussed and cited:
“The influence of microcrack width on the mechanical parameters in concrete with the addition of fly ash: Consideration of technological and ecological benefits”, Construction and Building Materials, 2019.
“An assessment of microcracks in the Interfacial Transition Zone of durable concrete composites with fly ash additives”, Composite Structures, 2018.
Reply:
We have discussed and cited other modern methods to reduce the crack width. Please check lines 54-57 in the revised manuscript.
Thank you again!
Best wishes!
Xingyan Shang, Jiangtao Yu, Lingzhi Li and Zhoudao Lu.

Round 2
Reviewer 3 Report
The authors have responded the major concerns. Since there is not sufficient enough for durability, the item 4 in the conclusion is meanless so far. Also, the new section 5 future studies is not proper and not sufficient as a section. The reviewer will suggest to place it in the conclusion as a short paragraph.
Author Response
Dear Editor,
Thank you very much for your valuable and kind recommendations. We have revised the paper according to your comments.
Question 1:
The authors have responded the major concerns. Since there is not sufficient enough for durability, the item 4 in the conclusion is meanless so far. Also, the new section 5 future studies is not proper and not sufficient as a section. The reviewer will suggest to place it in the conclusion as a short paragraph.
Reply:
We have deleted the item 4 of the conclusion in the previous version and placed the future studies in the conclusion. Please check lines 374-379 in the revised manuscript, which is highlighted in yellow.
Question 2:
Moderate English changes required.
Reply:
We have edited the English language and style of the manuscript. Please read and check.
Thank you again!
Best wishes!
Xingyan Shang, Jiangtao Yu, Lingzhi Li and Zhoudao Lu.

Reviewer 4 Report
The authors have addressed all the comments. I suggest the manuscript be published.
Author Response
Dear Editor,
Thank you very much for your suggestions. We have edited the English language and style of the manuscript. Please read and check.
Thank you again!
Best wishes!
Xingyan Shang, Jiangtao Yu, Lingzhi Li and Zhoudao Lu.
